# Spatial-Aware Feature Aggregation for Cross-View Image based Geo-Localization

**Yujiao Shi, Liu Liu, Xin Yu, Hongdong Li**

Australian National University, Canberra, Australia.
Australian Centre for Robotic Vision, Australia.
`{firstname.lastname}@anu.edu.au`

## Abstract

Recent works show that it is possible to train a deep network to determine the geographic location of a ground-level image (e.g., a Google street-view panorama) by matching it against a satellite map covering the wide geographic area of interest. Conventional deep networks, which often cast the problem as a metric embedding task, however, suffer from poor performance in terms of low recall rates. One of the key reasons is the vast differences between the two view modalities, *i.e.*, ground view versus aerial/satellite view. They not only exhibit very different visual appearances, but also have distinctive geometric configurations. Existing deep methods overlook those appearance and geometric differences, and instead use a brute force training procedure, leading to inferior performance. In this paper, we develop a new deep network to explicitly address these inherent differences between ground and aerial views. We observe that pixels lying on the same azimuth direction in an aerial image approximately correspond to a vertical image column in the ground view image. Thus, we propose a two-step approach to exploit this prior. The first step is to apply a regular polar transform to warp an aerial image such that its domain is closer to that of a ground-view panorama. Note that polar transform as a pure geometric transformation is agnostic to scene content, hence cannot bring the two domains into full alignment. Then, we add a subsequent spatial-attention mechanism which brings corresponding deep features closer in the embedding space. To improve the robustness of feature representation, we introduce a feature aggregation strategy via learning multiple spatial embeddings. By the above two-step approach, we achieve more discriminative deep representations, facilitating cross-view Geo-localization more accurate. Our experiments on standard benchmark datasets show significant performance boosting, achieving more than doubled recall rate compared with the previous state of the art. Remarkably, the recall rate@top-1 improves from 22.5% in [5] (or 40.7% in [11]) to 89.8% on CVUSA benchmark, and from 20.1% [5] to 81.0% on the new CVACT dataset.

## 1  Introduction

Image based Geo-localization is referred to the task of determining the location of an image (known as a query image) by comparing it with a large set of Geo-tagged database images. It has important computer vision applications such as for robot navigation, autonomous driving, as well as way-finding in AR/VR applications.

In this paper, we study ground-to-aerial cross-view image based Geo-localization problem. To be specific, the query image is a normal ground-level image (e.g., a street view image taken by a tourist) whereas the database images are collections of aerial/satellite images covering the same (though

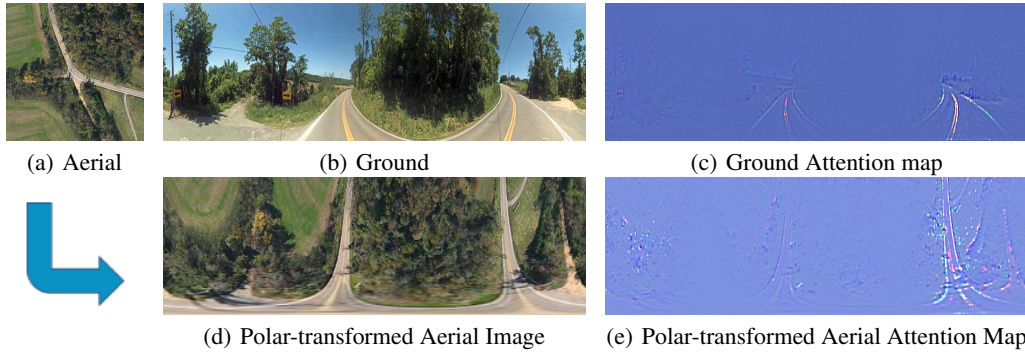

(a) Aerial&emsp;(b) Ground&emsp;(c) Ground Attention map

(d) Polar-transformed Aerial Image&emsp;(e) Polar-transformed Aerial Attention Map

Figure 1: Illustration of geometric correspondences between ground and aerial images, and visualization of our generated spatial embedding maps.

wider) geographic region. Cross-view image based localization is a very challenging task because the viewpoints (as well as imaging modality) between ground and aerial images are drastically different; their image visual appearances can also be far apart. As a result, finding feature correspondence between two views (even for a matching pair) can be very challenging. Recently, machine learning techniques (especially deep learning) have been applied to this task, showing promising results [5, 11, 19, 24].

Existing deep neural networks developed for this task often treat the cross-view localization problem as a standard image retrieval task, and are trained to find better image feature embeddings that bring matching image pairs (one from ground view, and one from aerial view) closer while pushing those unmatching pairs far apart. In other words, they cast the problem as a deep metric learning task, and thus learn feature representations purely based on image content (appearance or semantics) without taking into account spatial correspondences between ground and aerial views. To be precise, as seen in Figure 1(a) and Figure 1(b), one can easily observe that the locations of objects in an aerial image exhibit a strong spatial relationship with the ones in its corresponding ground image. Furthermore, the relative positions among objects also provide critical clues for the cross-view image matching.

By exploring such geometric configurations of the scenes, one can significantly reduce the ambiguity of the cross-view image matching problem, and this is the key idea of our paper, which will be described next.

Unlike conventional approaches, our method focuses on establishing spatial correspondences between these two domains explicitly and then learning feature correspondences from these two coarsely aligned domains. Although deep neural networks are able to learn any functional transformation in theory, explicitly aligning two domains based on geometric correspondences will reduce the burden of the learning process for domain alignment, thus facilitating the network convergence. In our method, we apply polar coordinate transform to aerial images, making it approximately aligned with a ground-view panorama, as shown in Figure 1(d). After polar transform, we train a Siamese-type network architecture to establish deep feature representation. Since polar transform does not take the scene content into account and the true correspondences between the two different domains are more complex than a simple polar transform, some objects may exhibit distortions. To remedy that, we develop a spatial attention based feature embedding module to extract position-aware features. Precisely, our spatial feature embedding module imposes different attention on different locations and then re-weights features to yield a global descriptor for an input image. In this manner, our method not only retains image content information but also encodes the layout information of object features. To achieve robustness of feature representation, we employ a feature aggregation strategy by learning multiple spatial feature embeddings and then aggregating the embedded features. We further employ a triplet loss to establish the feature correspondences between these cross-view images. Our extensive experimental results demonstrate that our method achieves superior Geo-localization performance to the state-of-the-art. Remarkably, the recall rate@top-1 improves from 22.5% in [5] (or 40.7% in [11]) to 89.8% on CVUSA benchmark, and from 20.1% [5] (or 46.9% in[11]) to 81.0% on the new CVACT dataset.

Contributions of this paper can be summarized as follows:

- We propose a new pipeline to address the cross-view Geo-localization problem. We first exploit the geometric correspondences between ground and aerial image domains to align these two domains explicitly by a polar transform, allowing the networks to focus on learning detailed scene-dependent feature correspondences.

- We present a spatial-aware attention module to re-weight features in accordance with feature locations. Since our method embeds relative positions between object features into image descriptors, our descriptors are more discriminative.

- We conduct extensive experiments which confirm that our proposed method significantly outperforms the state-of-the-art on two standard cross-view benchmark datasets. Our method achieves nearly 4-fold improvement in terms of top-1 recall, compared with the CVM-Net proposed in 2018 [5].

## 2  Related Work

Due to the drastic appearance and viewpoint changes, it is very difficult to match local features [12, 2, 18, 22] between ground and aerial images directly. Several methods [3, 10, 13] warp ground images into bird-view images and then match the warped images to the aerial ones. Jegou *et al.*[6] aggregate the residuals of local features to cluster centroids as image representations, known as VLAD descriptors. The work [17] aggregates a set of local features into a histogram, known as Bag of words, to attain a global descriptor. The aggregated descriptors are proved to be partially viewpoint and occlusion invariant, and thus facilitating image matching. However, hand-crafted features are still the performance bottleneck of traditional cross-view Geo-localization methods.

Deep neural networks have demonstrated their powerful image representation ability [14]. The seminal work [20] fine-tune AlexNet [8] on Imagenet [14] and Places [25] to extract features for the cross-view matching task. This work also indicates that the better discriminativeness of deep features compared to hand-crafted features. The work [21] fine-tunes CNNs by minimizing the feature distances between aerial and ground-view images and obtains better localization performance. [19] employs a triplet CNN architecture to learn feature embedding and achieves significant improvements. [5] embeds a NetVLAD layer on top of a VGG backbone network to represent the two-view images more discriminatively. Liu & Li [11] observe that orientations play a critical role in learning discriminative features. Thus, this method incorporates per-pixel orientation information into a CNN to learn orientation-selective features for the cross-view localization task. Shi *et al.*[15] propose a feature transport module to bridge the spatial and feature response domain differences between ground and aerial images. However, it might be difficult for networks to explore both geometric and feature correspondences simultaneously via a metric learning objective. Therefore, we propose to decouple the procedure of constructing geometric and feature correspondences, and let networks learn simple tasks.

## 3  Methodology

In this section, we first introduce the polar transform applied to aerial images for aligning these two cross-view domains, and then we present our spatial-aware position embedding module for descriptor extraction of both ground and aerial images. We employ a Siamese-like two-branch network architecture and our entire pipeline is illustrated in Figure 2.

### 3.1  Polar Transform

As we observed, pixels lying on the same azimuth direction in an aerial image approximately correspond to a vertical image column in the ground view image. Instead of enforcing neural networks to learn this mapping implicitly, we explicitly transform the aerial images and then roughly eliminate the geometric correspondence gap between these two domains. In doing so, we ease the task of learning multiple correspondences (*i.e.*, geometry and feature representations) and only need to learn a simple feature mapping task, thus significantly facilitating network convergence.

We apply polar transform to aerial images in order to build more apparent spatial correspondences between aerial and ground images. Specifically, we take the center of each aerial image as the polar origin and the north direction (as it is often available for a satellite image) as angle $0°$ in

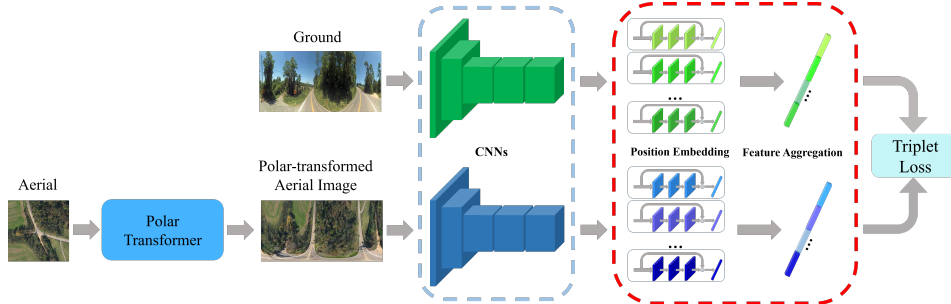

Figure 2: Illustration of the pipeline of our proposed method.

the polar transform. Note that there is no ad hoc pre-centering process for the aerial images, and we do not assume that the location of a query ground-level image corresponds to the center of an aerial image during testing. In fact, small offsets on the polar origin do not affect the appearance of polar-transformed aerial images severely, and the small appearance changes will be reduced by our SPE modules (as illustrated in detail in Section 3.2). On the contrary, when a large offset occurs, the aerial image should be regarded as a negative sample and the polar-transformed aerial image will be significantly different from the ground-truth one. In this manner, the polar transform effectively increases the discriminativeness of our model.

To facilitate training of our two-branch network, we constrain the size of the transformed aerial images to be the same as the ground ones $W_g \times H_g$. Note that, the size of the original aerial images is $A_a \times A_a$. Therefore, the polar transform between the original aerial image points $(x_i^s, y_i^s)$ and the target transformed aerial image points $(x_i^t, y_i^t)$ is defined as:

$$
\begin{aligned}
x_i^s &= \frac{A_a}{2} + \frac{A_a}{2} \frac{y_i^t}{H_g} \sin(\frac{2\pi}{W_g} x_i^t) \\
y_i^s &= \frac{A_a}{2} - \frac{A_a}{2} \frac{y_i^t}{H_g} \cos(\frac{2\pi}{W_g} x_i^t)
\end{aligned}
\tag{1}
$$

After polar transform, the objects in the transformed aerial images lie in similar positions to their counterparts in the ground images, as seen in Figure 1(d). However, the appearance distortions are still obvious in the transformed images because polar transform does not take the depth of the scene content into account. Reducing these distortion artifacts for image descriptor extraction is also desirable.

## 3.2 Spatial-aware Feature Aggregation (SAFA)

As illustrated in Figure 2, we first employ a backbone network, *i.e.*, the first sixteen layers of VGG19 [16], to extract features from ground and polar-transformed aerial images. Considering the features from aerial images undergo distortions, we intend to impose an attention mechanism to select salient features while suppressing the features caused by the distortions. Moreover, since spatial layout provides important clues for image matching, we aim to embed spatial configuration into our feature representation as well. Thus, we develop a spatial-aware feature aggregation (SAFA) module to alleviate the distortions in transformed aerial images while embedding the object features into a discriminative global image descriptor for image matching. Our SAFA is built upon the outputs of a Siamese network and learns to encode ground and aerial features individually. The detailed architecture of SAFA is shown in Figure 3.

**Spatial-aware Position Embedding Module (SPE):**

Our SPE is designed to encode the relative positions among object features extracted by the CNN backbone network, as well as the important features. In particular, given input feature maps from one branch, our SPE automatically determines an embedding position map from them. Note that, we do not enforce any additional supervision for SPE and it is learned in a self-attention fashion by a metric learning objective. Moreover, although polar transform can significantly reduce the domain gap in terms of geometric configuration, object distortions still exist and cannot be removed by an explicit

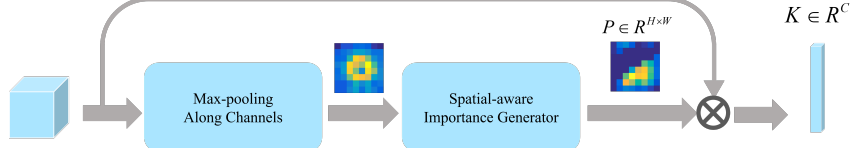

Figure 3: Spatial-aware position embedding module.

function. Thus, we employ SPE to select the features from transformed aerial images while reducing the impact of the distortion artifacts in the feature extraction.

Figure 3 illustrates the workflow of our SPE module. Our SPE first employs a max-pooling operator along feature channels to choose the most distinct object feature, and then adopts a Spatial-aware Importance Generator to generate a position embedding map. In the Spatial-aware Importance Generator, two fully connected layers are used to select features among the prominent ones as well as encode the spatial combinations and feature responses. In this manner, our method can mitigate the impacts of the features from distortions caused by polar transform while represent input images by using salient features. Furthermore, since we choose features based on a certain layout, the encoded features not only represent the emergence of certain objects but also reflect the positions of the objects. Hence, we encode the spatial layout information into feature representation, thus improving the discriminativeness of our descriptors.

Given the position embedding map $P \in R^{H \times W}$, the feature descriptor $K = \{k^c\}$, $c = 1, 2, ..., C$, is calculated as:

$$k^c = \langle \mathbf{f}^c, P \rangle_F, \tag{2}$$

where $\mathbf{f}^c \in R^{H \times W}$ represents the input feature map of the SPE module in the $c$-th channel, $\langle ., . \rangle$ denotes the Frobenius inner product of the two inputs, and $k^c$ is the embedded feature activation for the $c$-th channel.

As seen in Figure 1, only a certain region achieves high responses in the visualized feature maps. This indicates that our SPE not only localizes the salient features but also encodes the layout information of those features. Note that the SPE module is adopted in both the ground and aerial branches, and our objective forces them to encode correspondent features between these two branches.

**Multiple Position-embedded Feature Aggregation:** Motivated by the feature aggregation strategy [9], we aim to improve the robustness of our feature representation by aggregating our embedded features. Towards this goal, we employ multiple SPE modules with the same architecture but different weights to generate multiple embedding maps, and then encode input features in accordance with the different generated masks. For instance, some maps focus on the layout of roads while some focus on trees. Therefore, we can explore different spatial layout information in the input images. As illustrated in Figure 2, we concatenate the embedded features together as our final image descriptor. Note that, we do not impose any constraint on generating diverse embedding maps but learn embeddings through our metric learning objective. During training, in order to minimize the loss function, our descriptors should be more discriminative. Therefore, the loss function inherently forces our embedding maps to encode different spatial configurations to increase the discriminativeness of our embedded features.

### 3.3 Training Objective

We apply a metric learning objective to learn feature representations for both the ground and aerial image branches. The triplet loss is widely used to train deep networks for image localization and matching tasks [5, 11, 19]. The goal of the triplet loss is to make matching pairs closer while pushing unmatching pairs far apart. Similar to [5], we employ a weighted soft-margin triplet loss as our objective:

$$\mathcal{L} = \log(1 + e^{\gamma(d_{pos} - d_{neg})}), \tag{3}$$

where $d_{pos}$ and $d_{neg}$ are the $\ell_2$ distance of matching and unmatching image pairs. $\gamma$ is a parameter to adjust the gradient of the loss, thus controlling the convergence speed.

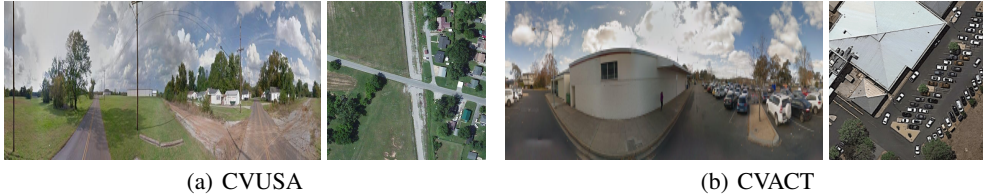

(a) CVUSA                                     (b) CVACT

Figure 4: Ground-to-aerial image pairs sampled from CVUSA [24] and CVACT [11]. Each subfigure illustrates a ground image **(Left)** and an aerial image **(right)**.

# 4 Experiments

**Training and Testing Datasets:** Our experiments are conducted on two standard benchmark datasets: CVUSA [24] and CVACT [11], where ground images are panoramas. CVUSA and CVACT are both cross-view datasets, and each dataset contains $35,532$ ground-and-aerial image pairs for training. CVUSA provides $8,884$ image pairs for testing and CVACT provides the same number of pairs for validation (denoted as CVACT_val). Besides, CVACT also provides $92,802$ cross-view image pairs with accurate Geo-tags to evaluate Geo-localization performance (denoted as CVACT_test). CVACT_test is a real geo-localization/retrieval test set where all aerial images within 5 meters to a query ground image are regarded as ground truth correspondences for this query image. In other words, for a query ground image, there may exists several corresponding aerial images in the database. Note that in these two datasets the ground and aerial images are captured at different time. Figure 4 presents sampled image pairs from these two datasets.

**Implementation Details:** We use the VGG16 model with pretrained weights on Imagenet [4] as our backbone to extract features from cross-view images, and the output of the last convolutional layer of VGG16 is fed into the proposed SAFA block[1]. The parameters in our proposed SPE module are randomly initialized. Similar to [5, 11], we set $\gamma$ to 10 for the triplet loss. Our network is trained with Adam optimizer [7], and the learning rate is set to $10^{-5}$. Exhaustive mini-batch strategy [19] is utilized to create triplet images within a batch, and the batch size $B_s$ is set to 32. In a mini-batch, there is 1 matching/positive aerial image and $B_s - 1$ unmatching/negative aerial images for each ground-view image. Thus, we construct $B_s(B_s - 1)$ triplets in total. Similarly, for each aerial image, there is 1 matching ground-view image and $B_s - 1$ unmatching ground-view images, and thus $B_s(B_s - 1)$ triplets are also constructed. Hence, we have $2B_s(B_s - 1)$ triplets in total within each batch.

**Evaluation Metric:** Similar to [19, 5, 11], we use the recall accuracy at top K as our evaluation metric to exam the performance of our model and compare with the state-of-the-art methods. Specifically, given a ground-level query image, it is regarded as "successfully localized" if its ground-truth aerial image is within the nearest top K retrieved images. The percentage of query images which have been correctly localized is reported as r@K.

## 4.1 Comparison with State-of-the-Art Methods

We compare our method with two recent state-of-the-art cross-view localization methods: CVM-NET [5] and Liu & Li's method [11]. For fair comparisons, we use the released models or codes provided by the authors. In our method, we apply polar transform to the aerial images and our SAFA outputs 8 spatial-aware embedding maps and then aggregate these embedded features, denoted as Polar_SAFA ($M = 8$). Note that, the dimension of our descriptors is as the same as that used in CVM-NET. We report recalls at top-1, top-5, top-10, up to top 1%, and the results are listed in Table 1.

As indicated by Table 1, our method significantly outperforms all the state-of-the-art methods. In particular, we almost **double** the recall at top-1 compared to Liu *et al.*'s method. The complete recall@K performance is shown in Figure 5.

Table 1: Comparison with state-of-the-art methods on CVUSA [24] and CVACT_val dataset [11].

| | CVUSA | | | | CVACT_val | | | |
|---|---|---|---|---|---|---|---|---|
| | r@1 | r@5 | r@10 | r@1% | r@1 | r@5 | r@10 | r@1% |
| CVM-NET [5] | 22.53 | 50.01 | 63.19 | 93.52 | 20.15 | 45.00 | 56.87 | 87.57 |
| Liu & Li [11] | 40.79 | 66.82 | 76.36 | 96.08 | 46.96 | 68.28 | 75.48 | 92.01 |
| Our polar-SAFA(M=8) | **89.84** | **96.93** | **98.14** | **99.64** | **81.03** | **92.80** | **94.84** | **98.17** |

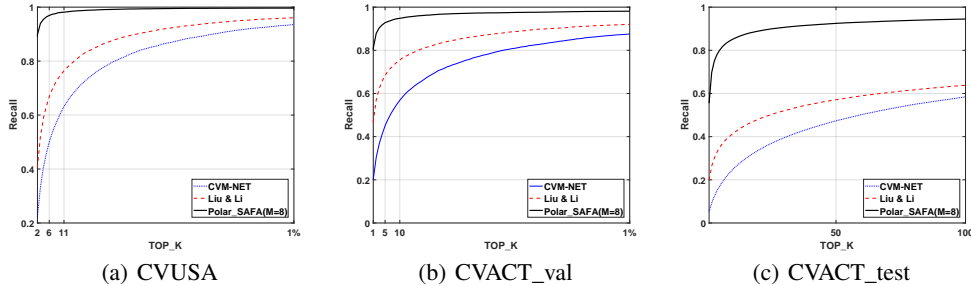

(a) CVUSA      (b) CVACT_val      (c) CVACT_test

Figure 5: Recall rates on cross-view Geo-localization datasets. This figure demonstrates that our method (*i.e.*, Polar_SAFA($M = 8$)) significantly outperforms the state-of-the-art methods.

## 4.2 Accurate Geo-localization

We conduct experiments on the large-scale CVACT_test dataset [11] to illustrate the effectiveness of our method for accurate city-scale Geo-localization applications. We also compare with the state-of-the-art methods, CVM-NET [5] andLiu & Li's method [11]. The recall performance at top-K is shown in Figure 5(c). Our method significantly outperforms the second-best method [11], with a relative improvement of $35.6\%$ at top-1.

## 4.3 Visualization of Learned Spatial Correspondences

To visualize our generated embedding maps, we employ the method of [23] to back-propagate the embedding maps to the input ground image as well as the polar-transformed aerial image. As visible in Figure 6, our SPE is able to encode similar spatial layout as well as feature correspondences between ground and polar-transformed aerial images. Furthermore, different SPE modules can generate different spatial embedding maps. In this way, we can encode multiple spatial layouts into our feature representations.

## 4.4 Ablation Study

In this part, we demonstrate the effectiveness of our proposed polar transform and Spatial-aware Position Embedding (SPE) modules. For the baseline network, we remove the polar transform from our network and replace the SPE module with a global max-pooling operator, which has been widely adopted in image retrieval tasks[5, 11, 1]. In this case, spatial correspondences between ground and aerial branches are not used and the baseline network is only trained by our triplet loss.

**Effects of Polar Transform:** To demonstrate the effectiveness of polar transform for the cross-view Geo-localization problem, we train our baseline network in two different settings: one takes original cross-view ground and aerial images, marked as VGG_gp, and the other takes ground and polar-transformed aerial images, marked as Polar_VGG_gp. As indicated in Table 2, applying polar transform to aerial images improves the performance greatly on both datasets.

Moreover, we also investigate the applicability of polar transform to other cross-view Geo-localization models.Liu & Li [11] needs an additional pixel-wise orientation map for input images and the orientation maps are not available for polar transformed images. Thus, we only conduct experiments on CVM-NET [5]. As illustrated in Table 2, using the polar-transformed aerial images as input, we even improve the performance of CVM-NET by $27.47\%$ on CVUSA and $14.77\%$ on CVACT at r@1.

| Ground | Polar-transformed Aerial | Ground | Polar-transformed Aerial |

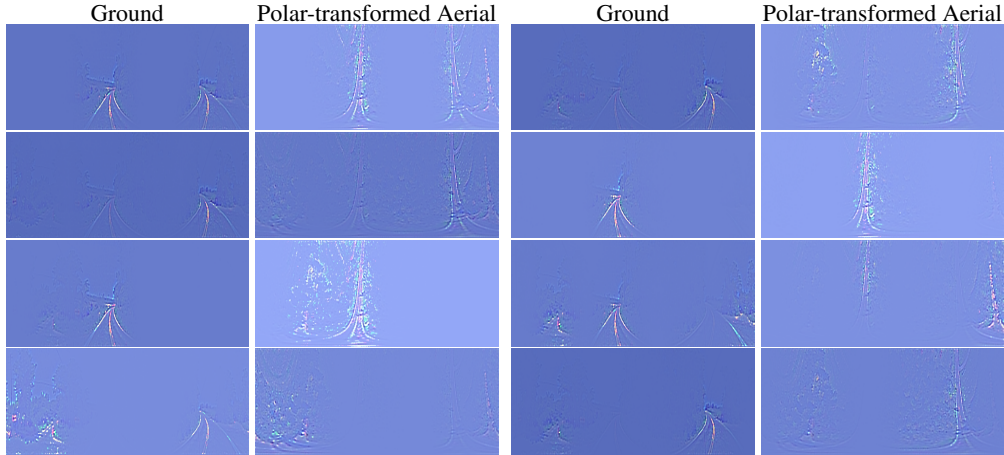

Figure 6: Visualization of eight-groups generated spatial embedding maps for ground and polar-transformed images. The corresponding ground and polar-transformed aerial images are shown in Figure 1(b) and Figure 1(d). (Best viewed on screen with zoom-in)

Table 2: Effectiveness demonstration of polar transform.

|  | CVUSA | | | | CVACT_val | | | |
|---|---|---|---|---|---|---|---|---|
|  | r@1 | r@5 | r@10 | r@1% | r@1 | r@5 | r@10 | r@1% |
| VGG_gp | 39.72 | 66.91 | 77.49 | 96.38 | 32.22 | 59.08 | 69.41 | 91.85 |
| Polar_VGG_gp | 65.74 | 84.76 | 89.91 | 98.30 | 56.65 | 79.20 | 84.98 | 95.76 |
| CVM-NET [5] | 22.53 | 50.01 | 63.19 | 93.52 | 20.15 | 45.00 | 56.87 | 87.57 |
| Polar_CVM-NET | 50.00 | 77.22 | 85.13 | 97.82 | 34.92 | 61.74 | 71.05 | 91.78 |

**Effects of Spatial-aware Position Embedding:**     We demonstrate the effectiveness of our proposed Spatial-aware Position Embedding (SPE) module using original cross-view images as inputs. We firstly replace the global max-pooling in VGG_gp with a single SPE module. Since our SPE module explicitly establishes spatial correspondences for cross-view images, it outperforms VGG_gp as indicated in Table 3. Especially, our single SPE model achieves 58.79% on CVUSA and 42.96% on CVACT_val for r@1, and obtains 48% and 33% relative improvements compared with VGG_gp, respectively.

**Effects of Multiple Spatial-aware Position Embeddings:**     To demonstrate the effectiveness of aggregating feature embeddings by using multiple SPE modules, we use different numbers of SPE modules, *i.e.*, 1, 2, 4, and 8, and report the recall rates in Table 3. The results indicates that as $M$ increases, we can obtain better recall performance. Note that, significant improvements ( 10%) for r@1 are obtained when $M$ increases from 1 to 2 and from 2 to 4. However, when $M$ increases from 4 to 8, we only attain slight improvements (<4%). Therefore, we do not increase $M$ to an even larger number. As indicated by Table 3, our method, combining polar transform and multiple SPE modules, achieves the best performance on both datasets. By employing polar transform, we improve the performance over 7%, thus demonstrating the effectiveness of polar transform as well.

## 5   Conclusion

We have proposed a new deep network to solve the cross-view image based Geo-localization problem. Our network addresses the difficulty caused by significant domain differences between ground-level and aerial-view images by a two-step procedure. The first step approximately brings the two image domains into a rough geometric alignment, and a subsequent spatial-attention mechanism further alleviates content-dependent domain discrepancy. Our key idea is to exploit available problem-dependent geometric priors of the task. In contrast to existing methods, we exploit the geometric constraint to coarsely align one domain to the other first. By doing so, we can force our network to focus on learning discriminative features without requiring to minimize the domain gap. Moreover,

Table 3: Effectiveness demonstration of the proposed SPE modules.

| | CVUSA | | | | CVACT_val | | | |
|---|---|---|---|---|---|---|---|---|
| | r@1 | r@5 | r@10 | r@1% | r@1 | r@5 | r@10 | r@1% |
| VGG_gp | 39.72 | 66.91 | 77.49 | 96.38 | 32.22 | 59.08 | 69.41 | 91.85 |
| SAFA ($M = 1$) | 58.79 | 84.19 | 90.84 | 99.08 | 42.96 | 71.51 | 80.56 | 95.48 |
| SAFA ($M = 2$) | 69.33 | 89.01 | 93.52 | 99.31 | 58.98 | 82.86 | 88.46 | 97.13 |
| SAFA ($M = 4$) | 79.93 | 93.29 | 96.15 | 99.54 | 74.61 | 90.02 | 93.03 | 98.01 |
| SAFA ($M = 8$) | 81.15 | 94.23 | 96.85 | 99.49 | 78.28 | 91.60 | 93.79 | 98.15 |
| Polar_SAFA ($M = 8$) | **89.84** | **96.93** | **98.14** | **99.64** | **81.03** | **92.80** | **94.84** | **98.17** |

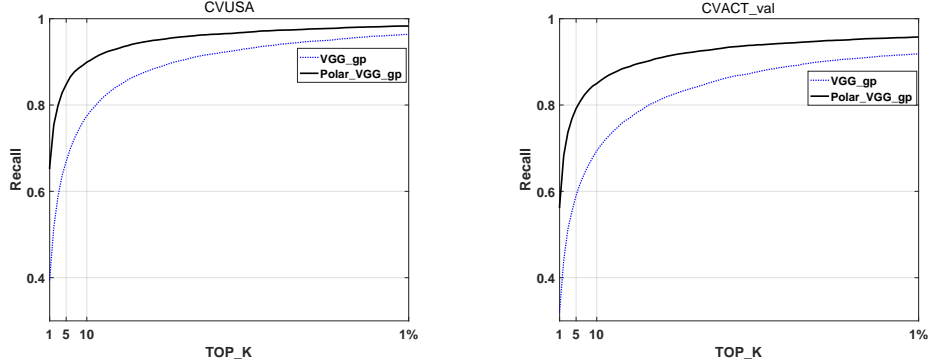

(a) With and without polar transform

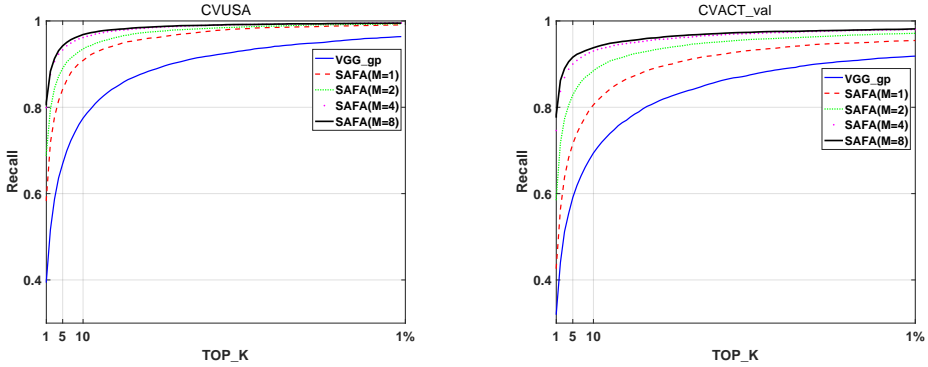

(b) Different number of SPE modules

Figure 7: Comparison of recalls on CVUSA [5] and CVACT_val [11] datasets.

we propose a spatial-aware feature aggregation module to not only embed features but also the feature layout information, achieving more discriminative image descriptors. Since the cross-view feature learning process has been decoupled, the domain gap does not affect feature learning. Our method is able to learn more discriminative image descriptors and thus outperforms the state-of-the-art. Although our current experiments are conducted on query ground images which are panoramas with known orientation, this restriction can be relaxed under the same network architecture and this is left as our future extension.

## Acknowledgments

This research is supported in part by China Scholarship Council (201708320417), the Australia Research Council ARC Centre of Excellence for Robotics Vision (CE140100016), ARC-Discovery (DP 190102261) and ARC-LIEF (190100080), and in part by a research gift from Baidu RAL (ApolloScapes-Robotics and Autonomous Driving Lab). The authors gratefully acknowledge the GPU gift donated by NVIDIA Corporation. We thank all anonymous reviewers for their constructive comments.

## Footnotes

[1]The code of this paper is available at https://github.com/shiyujiao/SAFA.

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
