[Supplementary Material]

# Spatial-Aware Feature Aggregation for Cross-View Image based Geo-Localization

## 1 Additional Visualization of Learned Spatial Correspondences

We provide additional visualizations of our generated spatial embedding maps between ground and polar-transformed aerial images in Figure 2, and the original matching pairs of the aerial and ground images as well as the corresponding polar-transformed aerial images are shown in Figure 1. As visible in Figure 2, our SPE is able to generate multiple different position embeddings for ground and aerial images, and the embeddings between these two domains are also consistent.

(a)

(b)

(c)

Figure 1: The original matching pairs of aerial and ground images as well as the corresponding polar-transformed aerial images. In each of the subfigure: **Left:** aerial image; **Middle:** ground image; **Right:** polar-transformed aerial image.

## 2 Generalization Ability of Polar Transform

To demonstrate the polar transform's effectiveness and generalization ability to other networks on this cross-view Geo-localization task, we retrain the CVM-NET with polar-transformed aerial images as input. Table 1 presents the results. With the polar transform, CVM-NET achieves higher recall performance on both CVUSA and CVACT_val datasets.

Figure 2: Visualization of our generated spatial embedding maps for ground and polar-transformed aerial images.

Table 1: Effectiveness demonstration of polar transform.

|  | CVUSA | | | | CVACT_val | | | |
|---|---|---|---|---|---|---|---|---|
|  | r@1 | r@5 | r@10 | r@1% | r@1 | r@5 | r@10 | r@1% |
| CVM-NET | 22.53 | 50.01 | 63.19 | 93.52 | 20.15 | 45.00 | 56.87 | 87.57 |
| Polar_CVM-NET | 50.00 | 77.22 | 85.13 | 97.82 | 34.92 | 61.74 | 71.05 | 91.78 |