[Reviews · NeurIPS 2019]

Reviewer 1



The main problem of the paper is that the contributions are somewhat incremental and the details of one of the main contributions in Section 3.2 are not as clear as they should be. For example, it is unclear to me what is the operation denoted by <.,.> in Equation 2. Is it elementwise multiplication or something else? Apparently there can be several "embedding maps M" but it is not clear to me how they differ.

Reviewer 2



# Paper structure - the paper is clearly written and well structured. Although I have the impression that if the reader is not fully aware of the SOTA in this subdomain, or what is standard or what is not, the current SOTA literature review is not enough. - Some parts need additional explanation, and I will detail below where. In general, I have the feeling that many details have not been addressed carefully enough leaving few doubts about the contribution. # General comments - The paper makes an interesting contribution, and mainly because, as stated in the introduction, i) a polar transform is applied to the database-retrieved aerial image to be matched and ii) the spatial attention module. However, these two aspects in conjunction with a non crystal clear experimental setup raise some questions. - in the polar transform, what are the coordinates at which the aerial image is transformed, and how are those chosen? Are all possible coordinates tested against the ground level image? L60-63 seem to clearly point to the fact that polar transform is applied _before_ the deep metric learning, and therefore the location is not learned jointly. - How are the aerial images sampled in terms of size and coverage? Is the GPS of the ground image used for a ballpark cropping? How is orientation and alignment chosen? This question arises naturally and I think that depending on the answer further comments might happen, e.g. whether the prior establishment of some of these parameters make the results realistic or not. E.g. in the data used, the pairs showed in figures seem to be nicely pre-aligned, such that the center of the aerial image corresponds to the position of the ground level image. Although this is fine when training, when testing, one does not have this amount of knowledge and images should not be pre-aligned, if not, there is no point in stating that the method is useful for localization. There is a full question mark about the retrieval part, which I hope will be clarified in the rebuttal. On the same line, datasets and experimental setups should be presented a bit in more detail. This point and the one before, make me wonder on whether results are very accurate only because the problem is simplified greatly not direclty by the polar transform, but by the implicit massive reduction of potential false positive rates. I expect a setup like [18], but it does not seem the case? - The development of the spatial attention module seems a bit arbitrary. Depending on the number of neurons, input images, amount of detail, clutter, etc. A full max-pooling along channels might well result in non-discriminative masking. L170-171 seem to be the main justification of this doing, and it sounds extremely arbitrary, which make the statement at L172-173 a pure hypothesis at this point. From the images in the results, although being able to focus on some features, these seem not to be really that discriminative, as e.g. in Fig 5 left the non-polar transfromed image seem to detect both streets (or the same one just distorted), while the polar transformed image activation seem only to highlight one. I probably miss something here, but it would be nice to have clarification on what is doing what, and, mostly, why. - L175-176 states that "we make SPE generate multiple embeddings ... ". How is this achieved? Intuitively, I'd say that several polar transformations are applied, and then features concatenated to perform matching. Still, in what written in the paper this does not seem the case as polar transformation and SPE seem independent modules. This further strenghten my question about how is the polar transformation applied in practice, and how are target orientation and coordinates chosen. - Would this pipeline really work for natural ground-level images, or panoramas would be needed because of the polar transform? Can this be tested somehow? - What is the level of "prior" matching needed from aerial images and ground images, in terms of content? Is the method robust to changes? Aerial images are verly likely to contain outdated information, can this be tested somehow by artificially changing content of current dataset? (maybe once the method is trained?) - Experiments are done properly and ablation studies are good, I just wonder the actual retrieval / matching problem how it influences the final alignment, e.g. if instead of small pre-centered image, only very large and coarse resolution aerial images are available, how the method scales to testing all possible locations for the polar transforms, etc - I wonder if other competitors could be added to the evaluations of another dataset used, since for now this does not really helps in positioning the contribution in SOTA. i see big improvements over the previous SOTA, but it is still hard to figure it out. Since such improvements are so big, and most of the time such massive improvements are suspicious, I'd be careful in motivating even further. Ablations really help in that direction, but might be not enough. # Specific comments - in the introduction, I'd explicitly state very clearly how the pipeline is applied at test time as well, from the selection of candidates to the final scoring. This information is missing also in the experimental section. - L171-173 need to be clarified - Section 4.datasets. data should be described a bit in more detail, although are freely accessible. the fact that so many pairs are established, it seems that aerial images are pre-cropped and pre-centered, which, if true, would invalidate the results in my opinion, because they would be unrealistic. I understand the competitors might evaluate in the same terms and settings, but the final numbers would still be unrealistically high for real world, because the dataset is unrealistic. - Tab 3 and L269: M is used as "the position embedding map" and cannot be used now to identify counts

Reviewer 3



On the positive side, the paper makes two interesting and effective technical contributions in the polar transform and the spatial-aware feature aggregation. Detailed experiments show that each of the two contributions alone is sufficient to obtain state-of-the-art results and that their combination further improves performance. I especially like the polar transform as it is a neat idea that is simple to implement, requires no parameter tuning / learning, and can be used as a pre-processing step to boost the performance of previous work (as shown in the supplementary material, where Polar CVM-Net (which uses the polar transform before applying CVM-Net [5]) already reaches state-of-the-art performance compared to standard CVM-Net and [12] on both CVUSA and CVACT_val). Given that cross-view localization between ground-level and aerial images is a challenging problem, I am surprised and very impressed by how much the proposed contributions improve performance. The paper clearly advances the state-of-the-art by a significant margin. Both the polar transform and the spatial-aware feature aggregation are technically sound. I am not aware of previous work on cross-view localization that uses either of them. The paper does a good job in terms of motivating its technical contributions, clearly explaining the need for both of them. The very detailed experimental evaluation further verifies the two contributions by showing the impact of each one individually and combined. The references to prior work seem adequate (although [Regmi, Borji, Cross-View Image Synthesis using Conditional GANs, CVPR 2018] and [Vo et al., Revisiting IM2GPS in the Deep Learning Era, ICCV 2017] also seem relevant) and the paper does a good job of explaining the differences of the proposed approach to previous methods. My main point of criticism is the clarity of presentations of parts of the paper, which I believe could be improved: 1) I found the description of the SPE module rather confusing. Looking at Eq. 2, it seems that embedding map M is strongly tied to the input feature map (basically selecting the maximum activation over all channels at a given pixel position). In particular, no additional learning seem to be involved, an impression that is corroborated by Fig. 3. If this is the case, I do not see how the embedding map provides further information that is not present in the input feature maps from Fig. 3. In contrast, Fig. 2 suggests that the SPE module performs a set of convolutions to obtain the embedding map M and then combining this spatial layout with the original input features. This would make more sense to me as I would not see how one would obtain benefits from multiple SPE modules without introducing additional trainable parameters. 2) No information is provided which layer of the VGG16 architecture is used as input to SAFA. If SAFA contains trainable information, then the architecture of the SPE modules used should be specified as well. Otherwise, it will be very hard to replicate the results reported in this paper (unless source code is released). In addition, it is unclear to me whether the weights of the Siamese network from Fig. 2 are shared between aerial and ground-level images. The color coding used in the figure seems to indicate that the parameters are not shared, which would make sense given that the two image sources are not geometrically aligned. 3) In order to produce similar descriptors for warped aerial images and ground-level panoramas, they would need to be aligned. There is a potential rotation ambiguity between the original panoramas and those obtained via the polar transform. However, I do not see how this would be handled by SAFA. Does the paper assume that the transformed aerial image has the same orientation as the original panorama, e.g., that the center of the panorama corresponds to the north direction? 4) Splitting the results on CVUSA and CVACT_val over multiple tables (Tab. 1, 2, and 3) makes it hard to directly compare the different variants of the proposed approach with state-of-the-art results. Combining all results into a single table should make the presentation of the results clearer to read. It should also open up enough space to include the results for Polar CVM-Net from the supp. mat. (which I think add quite some value to the paper by showing that the polar transformation can be used for other approaches as well). 5) Sec. 4.2 states that the CVACT_test set is used for evaluation while the tables only mention the CVACT_val set. Which one is used? 6) [15] (VGG) seems to be the wrong reference for the statement: "[15] aims to learn image descriptors that are invariant against large viewpoint changes." In general, it would be good to cite the conference versions of papers (e.g., [15] was published at ICLR 2015) rather than their arXiv versions. While the shortcomings listed above impact the reproducibility of the paper under review, and thus decrease its potential impact, I do not think that they are severe enough to justify a rejection of the paper as they can be addressed in a potential camera ready version of the paper. As such, I am recommending to accept the paper. ----- update after rebuttal and discussion ---- After reading the other reviews and the rebuttal, I still believe that the paper is worth to be accepted. While I had hoped for some more details regarding SAFA (I agree with R1 that this part is not really described in detail), I am satisfied with the rebuttal and trust the authors to provide more details in the final version of the paper. However, given the lack of technical details in the rebuttal on SAFA, I am reluctant to raise my initial score. Regarding the concern of a prior on the orientation alignment raised by R2: I also have the impression that the orientation (north direction) of both ground and aerial images seems to be roughly known. I don't think this is much of a limiting assumption though. For the aerial images, it should be possible to get a good estimate of north from other sensors. For ground level images, one could just use multiple orientation hypotheses at test time (for panoramas, this would simply be a shift of the center of the images). While this would come at the price of increased run-time, I don't think that this would be a problem in practice, especially given the significant increase in performance by the proposed approach. However, I think this part should be made clearer.

[Author Response · NeurIPS 2019]

**Reviewer #1:**

We thank R1 for confirming our contributions. As requested, we further clarify SAFA in Sec.3.2 as follows:

In SAFA, multiple embedding maps M are generated by the spatial-aware position embedding (SPE) modules. For each SPE module, the purpose of max-pooling along channels is to obtain the most dominant features across spatial locations, and the two fully-connected layers are used to select features among the prominent ones as well as encode the spatial information and responses of the chosen features. By doing so, the SPE is able to highlight salient features while mitigating the feature distortions between the cross-view images. Moreover, SPE modules are adopted in both the ground and aerial branches, and our objective forces them to encode corresponding features between these two branches. We employ multiple SPE modules and aggregate encoded features to increase the robustness and discriminativeness of our model. In doing so, our SPE modules are initialized with different weights and thus able to generate different attention maps. Figure 2 in the supplementary material provides the visualization of the generated different embedding maps. $\langle .,. \rangle$ denotes the Frobenius inner product of the two inputs.

**Reviewer #2:**

We thank R2 for the thorough and comprehensive comments. We are glad that R2 thinks the contribution is interesting.

**1. Polar coordinate.** The polar transform takes the center of each aerial image as the origin without using any ad hoc pre-centering process. In general, the north direction of a satellite map is often available, and thus we use it as angle $0°$ in the polar transform. During testing, we do not assume the ground-to-aerial pair is perfectly aligned. In fact, small offsets on the polar origin do not affect the appearance of polar-transformed aerial images obviously, and the small appearance changes will be reduced by our SPE modules. On the contrary, when a large offset occurs, the aerial image should be regarded as a negative sample and the polar-transformed aerial image will be significantly different from the ground-truth one. In this manner, our polar transform effectively increases the discriminativeness of our model.

**2. Actual retrieval.** In Sec. 4.2, our experiment on CVACT_test is a real retrieval case. The aerial images in the database densely cover the city of interest. However, there is no guarantee that the location of the query ground-level image corresponds to the center of an aerial image. We still take the center of each aerial image as the polar origin, and Fig. 6(c) demonstrates the robustness and effectiveness of our method to unknown center offsets.

**3. Multiple embeddings in SPE.** Only one polar transform is applied to aerial images and the multiple embeddings are obtained by different SPE modules. Please refer to the response to R1 for the details of SPE.

**4. Robust to changes.** The aerial and the ground images in our experiments are obtained at different times [22, 12]. Therefore, our model is able to tolerate appearance changes. Once an updated satellite map is available, we can directly apply the trained model to re-extract the features of the database images and then perform the retrieval.

**5. Other competitors and datasets.** Cross-view image localization is a newly emerging problem and the two competitive algorithms included in our submission are the most advanced ones: the CVM-NET is published in CVPR2018 and the work of Liu *et al.*is published in CVPR2019. CVACT dataset is released recently in CVPR2019.

**6. Settings in [18], limitations to panoramas.** We test our algorithm on the dataset of [18] although it is not the case for our work. We achieve 71.5% on recall@1%, which is much higher than that of the original paper (59.9%). It also demonstrates that our proposed algorithm is not restricted to the panorama case.

**Reviewer #3:**

Thanks for R3's supportive comments and confirming our contributions. We response R3's specific concerns below:

**1. Structure of SPE.** Our SPE modules are trainable and we will clarify this in the revised version. Please refer to our response to R1 for more details.

**2. Overall framework.** SAFA takes the output of the last convolutional layer of VGG16 as input and the two branches do not share weights. We will release the source code soon.

**3. Orientation misalignments.** We convert rotational misalignments between ground and aerial images into translational shifts by the polar transform, thus facilitating CNNs to extract features. Moreover, since our SAFA embeds the relative spatial relationship between features instead of absolute locations, our method is insensitive to the misalignments. Our algorithm achieves 85% on CVUSA and 79% on CVACT at r@1 within $\pm 20°$ orientation perturbations.

**4. CVACT_val and CVACT_test.** Both CVACT_val and CVACT_test are test sets and their names are inherited from [12]. As CVACT_val is evaluated by the same metric as CVUSA, we list them in the same Tables. In CVACT_test, when an estimated location from aerial images is less than 5 meters to the ground-truth GPS, it will be considered as successful geo-localization. This is different from the evaluation metric employed in CVACT_val and CVUSA. Therefore, we report the results on CVACT_test separately.

[Meta-Review · NeurIPS 2019]

The initial scores for this paper were: 5: Marginally below the acceptance threshold. 4: An okay submission, but not good enough; a reject. 7: A good submission; an accept. The main critiques of the negative reviewers were lack of clarity and missing details. R2 had also doubts about novelty. The positive reviewer also pointed out to the many missing details but overall thinks the paper has interesting technical contributions validated by detailed experimental evaluation. The authors have provided the rebuttal. In the follow-up discussion the positive reviewer (R3) keeps their positive rating trusting the authors to provide some of the details still missing after the rebuttal in the final version of the work. R3 argues for accepting the work. R2 after reading the rebuttal and the discussion (see below) upgrades their score to 6: Marginally above the acceptance threshold. R1 in the discussion clarifies their position regarding novelty, thinks the paper is borderline, keeps their 5 rating, but is not against acceptance (see their comment below). The final scores for this work are: 5: Marginally below the acceptance threshold.  6: Marginally above the acceptance threshold. 7: A good submission; an accept. This is a borderline case. In the end AC is convinced by the positive arguments of R3 supported by R2 and recommends to accept this work. Input from R2 in the discussion after reading the rebuttal: “I updated my review according to the rebuttal. I think authors did indeed clarify a lot of doubts I had about results and their significance, and I think that the improvement over SOTA is pretty significant at this point. The architectures devised for the task seem to improve a lot, setting a new SOTA on a trending task. I think the community will be happy to discuss this contribution. I gladly welcome the added explanations in regarding the SAFA / SPE modules. “ Input from R1 in the discussion after reading the rebuttal: With the "somewhat incremental contribution" I was referring to the use of polar coordinate transform which as such is not new but the application to this particular problem is new (as far as I am aware). Also, the other key contribution (SAFA) was not very well explained. Nevertheless, I think that the improvements outlined in the author feedback will clarify this. I still think that this paper is a borderline, considering that NeurIPS is very selective in general and there are typically more good papers submitted than can be accepted. As there is no borderline rating available I chose rating 5 originally but I am not against acceptance.